# Feasibility and acceptability of nationwide HPV vaccine introduction in Senegal: Findings from community-level cross-sectional surveys, 2020

Reena H. Doshi[1]*, Rebecca M. Casey[1,2], Nedghie Adrien[1,2], Alassane Ndiaye[3], Timothy Brennan[1,2], Jerlie Loko Roka[4], Awa Bathily[5], Cathy Ndiaye[6], Anyie Li[1,2], Julie Garon[1,2], Ousseynou Badiane[3], Aliou Diallo[7], Anagha Loharikar[1]

1 Global Immunization Division, Centers for Disease Control and Prevention, Atlanta, Georgia, United States Of America, 2 CDC Foundation, Atlanta, Georgia, United States Of America, 3 Expanded Programme on Immunization, Ministry of Health and Social Action, Dakar, Senegal, 4 Division of Global Health Protection, Centers for Disease control and Prevention, Dakar, Senegal, 5 Unicef, Dakar, Senegal, 6 PATH, Dakar, Senegal, 7 World Health Organization, Dakar, Senegal

* rdoshi@cdc.gov

**Data Availability Statement:** The data is the property of the Senegal Ministry of Health and Social Action. All data requests should be

## Abstract

In Senegal, cervical cancer is the most common cancer among women and the leading cause of morbidity and mortality from all cancers. In 2018, Senegal launched a national human papillomavirus (HPV) vaccination program with Gavi, the Vaccine Alliance (Gavi), support. HPV vaccination was incorporated into the national immunization program as a two-dose schedule, with a 6-12-month interval, to nine-year-old girls via routine immunization (RI) services at health facilities, schools and community outreach services throughout the year. During February to March 2020, we conducted interviews to assess the awareness, feasibility, and acceptability of the HPV vaccination program with a cross-sectional convenience sample of healthcare workers (HCWs), school personnel, community healthcare workers (cHCWs), parents, and community leaders from 77 rural and urban health facility catchment areas. Participants were asked questions on HPV vaccine knowledge, delivery, training, and community acceptability of the program. We conducted a descriptive analysis stratified by respondent type. Data were collected from 465 individuals: 77 HCW, 78 school personnel, 78 cHCWs, 152 parents, and community leaders. The majority of HCWs (83.1%) and cHCWs (74.4%) and school personnel (57.7%) attended a training on HPV vaccine before program launch. Of all respondents, most (52.5–87.2%) were able to correctly identify the target population. The majority of respondents (60.2–77.5%) felt that the vaccine was very accepted or accepted in the community. Senegal's HPV vaccine introduction program, among the first national programs in the African region, was accepted by community stakeholders. Training rates were high, and most respondents identified the target population correctly. However, continued technical support is needed for the integration of HPV vaccination as a RI activity for this non-traditional age group. The Senegal experience can be a useful resource for countries planning to introduce the HPV vaccine.

requested via email or phone to the immunization program: informatique@sante.gouv.sn or by Telephone: +221338694242.

**Funding:** This work was supported by Gavi, the Vaccine Alliance and titled "Evaluation of Human Papilloma Virus (HPV) Vaccine National Introduction in Low-and-Lower-Middle Income Countries" through a Contract No. ME 9422 12 20] awarded to CDC Foundation." The funders had no role in study design, data collection and analysis, decision to publish, or preparation of the manuscript".

**Competing interests:** The authors have declared that no competing interests exist.

## Introduction

Human papillomavirus (HPV) is the most common viral infection affecting the reproductive tract and is the primary cause of cervical cancer [1]. Each year 570,000 women are diagnosed, and 270,000 women die of HPV-associated cervical cancer globally. Most cervical cancer cases and deaths (>85%) occur in low resource settings, due to the lack of screening and treatment availability [1, 2].

Women in sub-Saharan Africa are disproportionately affected, with the highest incidence and mortality rates [3]. In 2018, over 20% of cancer deaths in sub-Saharan African women were because of cervical cancer [3]. In Senegal, cervical cancer is the most common cancer among women and the leading cause of morbidity and mortality for all cancers, with an estimated 1,876 new cases and 1,367 deaths each year [3].

In 2018, the World Health Organization (WHO) issued a call to action, with a global target of 90% of girls vaccinated with the HPV vaccine by 15 years of age to create a path for cervical cancer elimination worldwide [4]. Globally, HPV vaccination programs have led to reductions in vaccine-type HPV prevalence and in high grade lesions (cervical intraepithelial neoplasia, grade 2 or worse) among young women [5–7]. Since 2009, WHO has recommended HPV vaccination for primary prevention of HPV infection [1]. WHO recommends that all countries introduce the HPV vaccine into their national immunization program, with two doses for all girls aged 9–14 years, while continuing to strengthen cervical cancer screening and treatment. In 2017, WHO recommended vaccination of multiple cohorts (e.g., all girls aged 9–14 years) in the first year of introduction for the greatest public health impact, where feasible [1]. Despite these recommendations, fewer than 30% of the low- and middle-income countries have introduced HPV vaccination, compared with more than 85% of high-income countries [8]. As of October 2020, a total of 110 of 194 countries (57%) worldwide have integrated the HPV vaccine into their immunization programs [9].

Since 2013, following WHO recommendations, Gavi, the Vaccine Alliance (Gavi), has offered technical and financial support to eligible low-income countries for HPV vaccine introduction, beginning with small programs, to demonstrate the feasibility and understand how best to reach this novel target age group. From 2014–2016, Senegal successfully implemented a pilot project in two districts (Dakar Ouest and Meckhe) in the Dakar and Thiès regions [10, 11]. The pilot, a school based campaign, achieved high two-dose administrative coverage rate in both regions: 94% for HPV vaccine dose 1 and 92% for HPV vaccine dose 2 [10, 12]. Senegal subsequently launched a national HPV vaccination program at the end of 2018 with Gavi support. HPV vaccination was incorporated into the national immunization program with a two-dose schedule, using quadrivalent vaccine, given to nine-year-old girls (Cours Elémentaire 2 or grade 3) via routine immunization (RI), six months apart at health facilities (HF), schools, and community outreach services throughout the year [13, 14].

It can be challenging to reach young girls for vaccination because they do not routinely access health services. An evaluation of the Senegal pilot project revealed several key lessons learned, including the importance of sensitizing healthcare workers (HCWs) and community members, and the best strategies for delivering vaccines. However, there are gaps in our knowledge regarding how to expand the HPV vaccine delivery on a national scale. This evaluation aimed to understand the feasibility and acceptability of the HPV vaccine introduction in Senegal among key community stakeholders, including HCWs, community healthcare workers (cHCWs) delivering vaccines, school personnel, community leaders, and parents. Lessons learned will strengthen the HPV vaccination program in Senegal and help identify strengths and weaknesses, as well as guide program decision-making and implementation in similar settings throughout sub-Saharan Africa and worldwide.

## Methods

### Ethical approval

The protocol was reviewed by the US Centers for Disease Control and Prevention and received a non-research determination. Senegalese authorities considered this to be a program evaluation. Verbal informed consent was obtained from all participants in lieu of written consent due to the low literacy rate in Senegal and to ensure participants that the survey was completely anonymous. Consent was documented in the electronic survey tool.

### Sampling strategy

We conducted interviews with a cross-sectional sample of HCWs, cHCWs, school personnel, community leaders and parents to assess knowledge and awareness of HPV and the HPV vaccine, vaccine acceptability, and feasibility of the national HPV vaccination program in Senegal.

The sample included all regions, conducted concurrently with an economic evaluation of HPV vaccine introduction not described in this report. Detailed methods are described elsewhere [15]. Briefly, Senegal is composed of 14 regions and 77 districts. Districts were stratified into rural, urban, and mixed (districts including both urban and rural HFs categories based on the setting of HF. In total, 31 districts (four urban, 13 rural, and 14 mixed) were selected, using probability proportional to size (PPS) sampling with the volume of measles vaccine first doses (MCV1) delivered as the size variable. Within sampled districts, a minimum of two HFs were selected by simple random sampling (SRS). A total of 77 HFs (18 urban, 59 rural) were selected.

### Survey population

At each pre-selected HF, one HCW and one cHCW most closely associated with HPV vaccination were interviewed. The cHCW was asked to assist in identifying two parents, one community leader (e.g., political or religious leaders), and one primary school in the facility catchment area. From each school, at least one staff member closely involved in the HPV vaccination was interviewed (Fig 1).

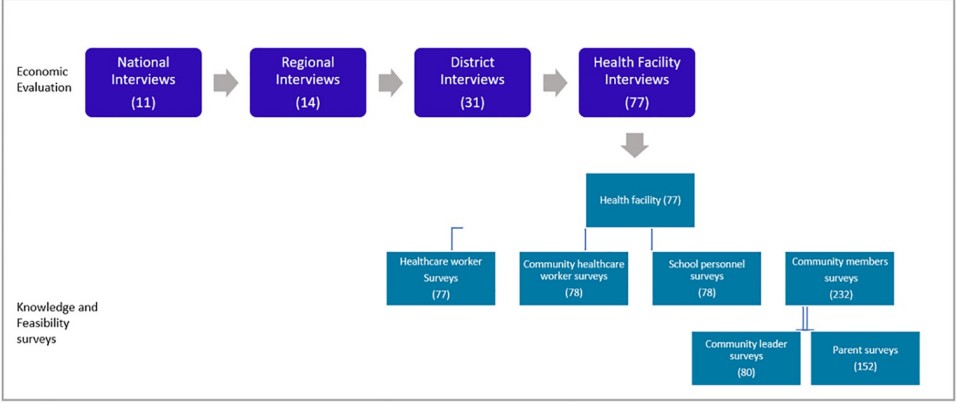

**Fig 1. Sampling strategy for the HPV vaccination program feasibility and acceptability surveys—Senegal, 2020.** HPV = human papillomavirus. Economic evaluation = This was a separate component of the evaluation and was designed to assess the cost of national introduction of HPV vaccine in a single-age cohort of nine-year-old girls and evaluate the costs of routine HPV vaccination in Senegal. It will be published separately.

## Survey instruments

The interviews were conducted by trained data collectors, using a standardized, structured questionnaire to better understand the feasibility of program implementation, the aspects of communication, training, social mobilization, and rumors about the HPV vaccine. Questionnaires were developed for each study population and included questions on demographics, HPV infection and the HPV vaccine, the feasibility of delivery strategy, target population, and vaccine acceptance. Questionnaires included closed-ended multiple choice questions on key training concepts for HCWs' acceptance and willingness to recommend the HPV vaccine, as well as open-ended questions on overall best practices, challenges, and HPV vaccination program recommendations. Questionnaires were initially developed in English and then translated to French; translation to the local language, Wolof, was done during data collection, as necessary. All questionnaires were piloted before data collection and were adapted, as needed.

## Data collection

Five supervisors and 11 interviewers were selected and attended a one-week training in Dakar. All data collectors were trained on tablet use and strategies for conducting the feasibility and acceptability interviews; then they attended a half-day field practice at a HF in Dakar. Data were collected using Open Data Kit (ODK) Collect and were uploaded daily with regular backups to the ODK aggregate server. Data collection took place between February 23–March 6, 2020.

## Data analysis

Data were exported from ODK into SAS version 9.4 (Cary, North Carolina, USA) and Microsoft Excel®. Descriptive analyses, frequencies, and percentages for qualitative variables and as median and interquartile range (IQR) for quantitative variables were completed for the demographics, training, workload, best practices, challenges, and program recommendations across each respondent group.

## Results

### Demographics

Data were collected from HFs in 31 districts and 465 individuals completed the interviews, including, 77 HCWs, 78 school personnel, 78 cHCWs, 152 parents and 80 community leaders. Among HCWs, 49.4% were female; the median age was 37 [Interquartile range (IQR): 35, 42], and 96.1% were nurses (Table 1). Among school personnel, 19.2% were female and the median age was 43.5 (IQR: 38, 50). Among school personnel, only teachers (41.0%) and directors (59.0%) were interviewed. Among cHCW, 65.4% were female and the median age was 40 (IQR: 33.5, 51). Among parents, 79.6% were female and the median age was 40 (IQR: 32, 47). Among community leaders, 68.8% were male and the median age was 56.5 (IQR:47.5, 65).

### HPV vaccination program service delivery

Community HCWs reported that girls could receive the HPV vaccine at health facilities (92.3%), school (85.9%), and in the community (60.3%) (Table 2). Most parents and community leaders indicated that girls could receive the vaccine at health facilities (86.1% and 90%, respectively) and schools (71.1% and 70.0%, respectively). Only a small proportion of parents and community leaders knew that girls could receive the vaccine in the community (28.3%, 33.8%).

**Table 1. Demographic characteristics by respondent group, HPV vaccination program feasibility and acceptability surveys—Senegal, 2020.**

| | Healthcare Workers (N = 77) | | School Personnel (N = 78) | | Community healthcare workers (N = 78) | | Parents (N = 152) | | Community leaders (N = 80) | |
|---|---|---|---|---|---|---|---|---|---|---|
| | n | % | n | % | n | % | n | % | n | % |
| **Sex** | | | | | | | | | | |
| Male | 39 | 50.7 | 63 | 80.8 | 27 | 34.6 | 31 | 20.4 | 55 | 68.8 |
| Female | 38 | 49.4 | 15 | 19.2 | 51 | 65.4 | 121 | 79.6 | 25 | 31.3 |
| **Age, years [median (IQR)]** | 37 (35–42) | | 43.5 (38–50) | | 40 (33.5–51) | | 40 (32–47) | | 56.5 (47.5–65) | |
| Under 25 | 0 | 0.0 | 1 | 1.3 | 5 | 6.4 | 6 | 3.9 | 0 | 0.0 |
| 25–34 | 19 | 24.7 | 13 | 16.7 | 17 | 21.8 | 39 | 25.7 | 3 | 3.8 |
| 35–44 | 43 | 55.8 | 28 | 35.9 | 23 | 29.5 | 56 | 36.8 | 12 | 15.0 |
| 45–54 | 9 | 11.7 | 32 | 41.0 | 18 | 23.1 | 31 | 20.4 | 19 | 23.8 |
| **Profession or Role** | | | | | | | | | | |
| Parent | - | - | - | - | - | - | 152 | 100 | 0 | 0.0 |
| Political leader | - | - | - | - | - | - | - | - | 10 | 12.5 |
| Religious leader | - | - | - | - | - | - | - | - | 5 | 6.3 |
| Administrative leader | - | - | - | - | - | - | - | - | 22 | 27.5 |
| Traditional healer | - | - | - | - | - | - | - | - | 1 | 1.3 |
| Influential community member | - | - | - | - | - | - | - | - | 42 | 52.5 |
| Teacher | - | - | 32 | 41.0 | - | - | - | - | - | - |
| School Director | - | - | 46 | 59.0 | - | - | - | - | - | - |
| Nurse | 74 | 96.1 | - | - | - | - | - | - | - | - |
| Midwife | 2 | 2.6 | - | - | - | - | - | - | - | - |
| cHCW | - | - | - | - | 67 | 85.9 | - | - | - | - |
| Community outreach worker "Badjenu Gox" | - | - | - | - | 9 | 11.5 | - | - | - | - |
| Other | 1 | 1.3 | - | - | 2 | 2.6 | - | - | - | - |

HPV = human papillomavirus.

Unreported values (-) indicate that data were not collected for that respondent group.

Most schools included in the sample offered HPV vaccination (80.8%) and of those, 81.0% provided the vaccine twice a year. Among these schools 44.4% identified the eligible population using a roster of girls by grade, while 42.9% used the girls' age. If an eligible girl was absent on the day of vaccination, 71.4% indicated that they would tell the child to visit a health facility for vaccination, while 19.0% would ensure that the child was vaccinated at the next school immunization session.

Less than half of HCWs (41.6%) and school personnel (42.3%) indicated that 50–80% of eligible girls were in school. The majority of HCWs (69.0%) and almost half of cHCWs (44.2%) indicated that community outreach workers identified girls that are not in school. Only a small number of HCWs (5.0%) and cHCWs (14.3%) indicated that they made no special effort to identify out- of- school girls, despite, 38.0% of HCWs and 19.5% of cHCWs indicating that it was difficult to identify these out- of- school girls.

About half of HCWs (58.0%) and cHCWs (41.6%) indicated that they used or would use public records to verify that girls are age-eligible for vaccination. Almost half (45.5%) of the HCWs indicated that they utilized data provided by the national statistics office to determine the target eligible population for the first dose vaccination; almost all (97.4%) HCWs stated that they determined eligibility for the second dose from the list of girls who received the first dose. Two-thirds of the HCWs (64.9%) stated their health facility did not have a microplan for

**Table 2. HPV vaccination program service delivery, HPV vaccination program feasibility and acceptability surveys—Senegal, 2020.**

| | Healthcare Workers (N = 77) | | School personnel (N = 78) | | Community healthcare workers (N = 78) | | Parents (N = 152) | | Community leaders (N = 80) | |
|---|---|---|---|---|---|---|---|---|---|---|
| | n | % | n | % | n | % | n | % | n | % |
| **Where can girls receive the HPV vaccine?** | | | | | | | | | | |
| School | - | - | - | - | 67 | 85.9 | 108 | 71.1 | 56 | 70.0 |
| Health facility | - | - | - | - | 72 | 92.3 | 131 | 86.2 | 72 | 90.0 |
| In the community | - | - | - | - | 47 | 60.3 | 43 | 28.3 | 27 | 33.8 |
| Do not know | - | - | - | - | 0 | 0.0 | 8 | 5.3 | 2 | 2.5 |
| **Is the HPV vaccine offered at this school?** | | | | | | | | | | |
| Yes | - | - | 63 | 80.8 | - | - | - | - | - | - |
| No | - | - | 15 | 19.2 | - | - | - | - | - | - |
| **How often is the HPV vaccine offered at school?**[*] | | | | | | | | | | |
| 2 times per year | - | - | 51 | 81.0 | - | - | - | - | - | - |
| <1 time per month (but more than 2 times per year) | - | - | 8 | 12.7 | - | - | - | - | - | - |
| 2 times a month | - | - | 0 | 0.0 | - | - | - | - | - | - |
| >3 times a month | - | - | 0 | 0.0 | - | - | - | - | - | - |
| Do not know | - | - | 4 | 6.3 | - | - | - | - | - | - |
| **What is your participation in the HPV vaccination program?**[*] | | | | | | | | | | |
| Fill out the data collection forms for schools (preparation of the list of eligible girls, enumeration of girls) | - | - | 32 | 50.8 | - | - | - | - | - | - |
| Organize education meetings for girls and parents | - | - | 40 | 63.5 | - | - | - | - | - | - |
| Keep the vaccination cards at the school | - | - | 14 | 22.2 | - | - | - | - | - | - |
| Other | - | - | 19 | 30.2 | - | - | - | - | - | - |
| **How do you determine the eligible population for HPV vaccination in this school**[*] | | | | | | | | | | |
| All girls from the correct grade | - | - | 28 | 44.4 | - | - | - | - | - | - |
| Dates/ Age | | | 27 | 42.9 | | | | | | |
| Other | - | - | 7 | 11.1 | - | - | - | - | - | - |
| **What would you do if eligible girl is absent the day of vaccination**[*] | | | | | | | | | | |
| Ensure that the child is vaccinated at the next school immunization session | - | - | 12 | 19.0 | - | - | - | - | - | - |
| Tell the girl to visit a health facility for vaccination | | | 45 | 71.4 | | | | | | |
| Nothing | - | - | 1 | 1.6 | - | - | - | - | - | - |
| Other | - | - | 5 | 7.9 | - | - | - | - | - | - |
| **What percentage of 9-year-old girls are in school?** | | | | | | | | | | |
| >80% | 25 | 32.5 | 33 | 42.3 | 44 | 56.4 | 101 | 66.4 | 45 | 56.3 |
| 50–-80% | 32 | 41.6 | 33 | 42.3 | 22 | 28.2 | 36 | 23.7 | 21 | 26.3 |
| 30–-50% | 11 | 14.3 | 7 | 9.0 | 11 | 14.1 | 5 | 3.3 | 9 | 11.3 |
| <30% | 5 | 6.5 | 4 | 5.1 | 0 | 0.0 | 3 | 2.0 | 2 | 2.5 |
| Do not know | 4 | 5.2 | 1 | 1.3 | 1 | 1.3 | 7 | 4.6 | 3 | 3.8 |
| **How do you identify girls that are not in school?** | | | | | | | | | | |
| No effort | 4 | 5.0 | - | - | 11 | 14.3 | - | - | - | - |
| Community outreach workers | 53 | 69.0 | - | - | 34 | 44.2 | - | - | - | - |
| Community leaders | 20 | 26.0 | - | - | 15 | 19.5 | - | - | - | - |
| Public records | 0 | 0.0 | - | - | 10 | 13.0 | - | - | - | - |
| Work with the parents | 10 | 13.0 | - | - | 15 | 19.5 | - | - | - | - |
| Other | 9 | 12.0 | - | - | 13 | 16.9 | - | - | - | - |
| Do not know | 1 | 1.0 | - | - | 5 | 6.5 | - | - | - | - |
| **Is it hard to identify girls?** | | | | | | | | | | |
| Yes | 29 | 38.0 | - | - | 15 | 19.5 | - | - | - | - |
| No | 47 | 61.0 | - | - | 62 | 80.5 | - | - | - | - |
| **How do you verify girls are age-eligible for the HPV vaccine?** | | | | | | | | | | |
| Public records | 45 | 58.0 | - | - | 32 | 41.6 | - | - | - | - |
| Ask the girl | 2 | 3.0 | - | - | 0 | 0.0 | - | - | - | - |

*(Continued)*

**Table 2.** (Continued)

| | Healthcare Workers (N = 77) | | School personnel (N = 78) | | Community healthcare workers (N = 78) | | Parents (N = 152) | | Community leaders (N = 80) | |
|---|---|---|---|---|---|---|---|---|---|---|
| | n | % | n | % | n | % | n | % | n | % |
| Ask the teacher | 14 | 18.0 | - | - | 16 | 20.8 | - | - | - | - |
| Ask the parents | 13 | 17.0 | - | - | 26 | 33.8 | - | - | - | - |
| Other | 3 | 4.0 | - | - | 3 | 3.9 | - | - | - | - |
| **Was there an impact on your workload?** | | | | | | | | | | |
| Yes | 40 | 51.9 | 20 | 25.6 | 26 | 33.3 | | | | |
| No | 37 | 48.1 | 58 | 74.4 | 52 | 66.7 | | | | |
| **How do you determine the eligible population for the HPV vaccine dose 1?** | | | | | | | | | | |
| Numbers from the national statistics office | 35 | 45.5 | - | - | - | - | - | - | - | - |
| Ministry of health (district level) data | 23 | 29.9 | - | - | - | - | - | - | - | - |
| Enumeration of girls in area served by the health facility | 7 | 9.1 | - | - | - | - | - | - | - | - |
| Health facility registers | 4 | 5.2 | - | - | - | - | - | - | - | - |
| School based records (Ministry of Education) | 14 | 18.2 | - | - | - | - | - | - | - | - |
| Birth records | 2 | 2.6 | - | - | - | - | - | - | - | - |
| Do not know | 7 | 9.1 | - | - | - | - | - | - | - | - |
| Other | 1 | 1.3 | - | - | - | - | - | - | - | - |
| **How do you determine the eligible population for the HPV vaccine dose 2?** | - | - | | | | | | | | |
| From the list of girls who received the first dose | 75 | 97.44 | - | - | - | - | - | - | - | - |
| Other | 3 | 3.9 | - | - | - | - | - | - | - | - |
| Do not know | 0 | 0.0 | - | - | - | - | - | - | - | - |
| **Does your health facility have a microplan?** | | | | | | | | | | |
| Yes | 27 | 35.1 | - | - | - | - | - | - | - | - |
| No | 50 | 64.9 | - | - | - | - | - | - | - | - |

*n = 63, represents schools that indicated they offer HPV vaccination

HPV = human papillomavirus.

Unreported values (-) indicate that data were not collected for that respondent group.

the HPV vaccine. The impact of introducing routine HPV vaccination on respondent workload varied; 51.9% of HCWs, 25.6% of school personnel, and 33.3% of community HCWs stated an impact.

## Knowledge of HPV and the HPV vaccine

Most HCWs (83.1%), cHCWs (74.4%), and over half of the school personnel (57.7%), attended a training on the HPV vaccine before the launch (Table 3). The majority of the HCWs (81.3%) and school personnel (57.8%) felt that the training was sufficient, while 48.3% of the community HCWs felt it was sufficient. Most HCWs (98.7%), school personnel (92.3%) and cHCWs (92.3%) correctly reported that HPV infection can cause cervical cancer. All HCWs, most (87.2%) cHCWs, and two-thirds of the school personnel (66.7%) knew that two doses of HPV vaccine are needed to complete the series. Almost all HCWs (94.8%), school personnel (87.2%), cHCWs (84.6%) and most of the parents (63.2%) and community leaders (52.5%) were able to identify the correct target population. A few HCWs (5.2%) and cHCWs (5.1%) identified multiple-age cohorts of 9–14 year-old girls as the target age eligibility. About one-third of the HCWs (36.4%) knew the contraindications for vaccination included a history of allergic reactions to vaccines (36.4%), pregnancy (25.7%), and, high fever or illness (32.5%).

**Table 3. Knowledge of HPV and HPV vaccine among key community stakeholders, HPV vaccination program feasibility and acceptability survey—Senegal, 2020.**

| | Healthcare Workers (N = 77) | | School Personnel (N = 78) | | Community Healthcare Workers (N = 78) | | Parents (N = 152) | | Community Leaders (N = 80) | |
|---|---|---|---|---|---|---|---|---|---|---|
| | n | % | n | % | n | % | n | % | n | % |
| **Attended a training (health workers, teachers, council leaders) or meeting (community leaders) on HPV vaccine** | | | | | | | | | | |
| Yes | 64 | 83.1 | 45 | 57.7 | 58 | 74.4 | - | - | - | - |
| No | 13 | 16.9 | 33 | 42.3 | 20 | 25.6 | - | - | - | - |
| **Was the training material sufficient?** | | | | | | | | | | |
| Yes | 52 | 81.3 | 26 | 57.8 | 28 | 48.3 | - | - | - | - |
| No | 12 | 18.8 | 19 | 42.2 | 26 | 44.8 | - | - | - | - |
| **What does HPV cause?** | | | | | | | | | | |
| Cervical Cancer | 76 | 98.7 | 72 | 92.3 | 72 | 92.3 | 107 | 70.4 | 62 | 77.5 |
| Other cancers | 3 | 3.9 | 1 | 1.3 | 1 | 1.3 | 5 | 3.3 | 3 | 3.8 |
| Genital warts | 5 | 6.5 | 0 | 0.0 | 1 | 1.3 | 0 | 0.0 | 0 | 0.0 |
| Do not know | 1 | 1.3 | 5 | 6.4 | 6 | 7.7 | 42 | 27.6 | 16 | 20.0 |
| Other (e.g, STIs, sterility) | 5 | 6.5 | 4 | 5.1 | 0 | 0.0 | 2 | 1.3 | 4 | 5.0 |
| Other | 76 | 98.7 | 72 | 92.3 | 72 | 92.3 | 107 | 70.4 | 62 | 77.5 |
| **Target population identified** | | | | | | | | | | |
| 9-year-old girls | 73 | 94.8 | 68 | 87.2 | 66 | 84.6 | 96 | 63.2 | 42 | 52.5 |
| 9-14-year-old girls | 4 | 5.2 | 0 | 0.0 | 4 | 5.1 | 0 | 0.0 | 0 | 0.0 |
| Do not know | 0 | 0.0 | 2 | 2.6 | 0 | 0.0 | 2 | 1.3 | 1 | 1.3 |
| Other (e.g., 9–15) | 0 | 0.0 | 0 | 0.0 | 0 | 0.0 | 26 | 17.1 | 18 | 22.5 |
| **Number of doses of vaccine recommended for immunocompromised** | | | | | | | | | | |
| 1 | 0 | 0.0 | 1 | 1.3 | 1 | 1.3 | 8 | 5.3 | 4 | 5.0 |
| 2 | 77 | 100.0 | 52 | 66.7 | 68 | 87.2 | 70 | 46.1 | 23 | 28.8 |
| 3 | 0 | 0.0 | 6 | 7.7 | 3 | 3.8 | 14 | 9.2 | 14 | 17.5 |
| Don't know | 0 | 0.0 | 19 | 24.4 | 5 | 6.4 | 53 | 34.9 | 36 | 45.0 |
| **Min interval between doses*** | | | | | | | | | | |
| 6 months | 77 | 100.0 | 48 | 82.8 | 64 | 88.9 | 58 | 63.7 | 20 | 48.8 |
| less than 3 months | 0 | 0.0 | 1 | 1.7 | 2 | 2.8 | 11 | 12.1 | 7 | 17.1 |
| Don't know | 0 | 0.0 | 6 | 10.3 | 2 | 2.8 | 16 | 17.6 | 12 | 29.3 |
| Other | 0 | 0.0 | 3 | 5.2 | 4 | 5.6 | 6 | 6.6 | 2 | 4.9 |
| **Contraindications** | | | | | | | | | | |
| History of allergic reactions to vaccines (components) | 28 | 36.4 | - | - | 6 | 7.7 | - | - | - | - |
| Pregnancy | 19 | 24.7 | - | - | 11 | 14.1 | - | - | - | - |
| Serious fever or illness | 25 | 32.5 | - | - | 15 | 19.2 | - | - | - | - |
| HIV | 4 | 5.2 | - | - | 0 | 0.0 | - | - | - | - |
| Don't know | 13 | 16.9 | - | - | 29 | 37.2 | - | - | - | - |
| Other (sexually active girls) | 34 | 44.2 | - | - | 31 | 39.7 | - | - | - | - |

*denominator is based on subset who indicated two or more doses

HPV = human papillomavirus.

Fewer cHCWs were aware of the contraindications; history of allergic reactions to vaccines (7.7%), pregnancy (14.1%), and, high fever or illness (19.2%).

## Vaccine acceptability, communication, and social mobilization

Overall, all respondent groups felt that there had been some community resistance since the program had started, but this varied by respondent group (HCWs 66.2%, school personnel 46.2%, cHCWs 51.3%, parents 9.2%, community leaders 8.8%) (Table 4). Most HCWs (84.4%), two-thirds of the school personnel (66.7%), and cHCWs (65.4%) reported hearing rumors about the HPV vaccine. Less than half (43.4%) of the parents and about half (52.5%) of the community leaders reported hearing such rumors.

**Table 4. Acceptability of HPV vaccine and social mobilization, HPV vaccination program feasibility and acceptability survey—Senegal, 2020.**

| | Healthcare Workers (N = 77) | | School Personnel (N = 78) | | Community Healthcare Workers (N = 78) | | Parents (N = 152) | | Community Leaders (N = 80) | |
|---|---|---|---|---|---|---|---|---|---|---|
| | n | % | n | % | n | % | n | % | n | % |
| **Encountered any resistance since the introduction of the vaccine? (e.g, by HCWs, teachers, parents)** | | | | | | | | | | |
| Yes | 51 | 66.2 | 36 | 46.2 | 40 | 51.3 | 14 | 9.2 | 7 | 8.8 |
| No | 26 | 33.8 | 35 | 44.9 | 38 | 48.7 | 124 | 81.6 | 70 | 87.5 |
| Do not know | 0 | 0.0 | 0 | 0.0 | 0 | 0.0 | 14 | 9.2 | 3 | 3.8 |
| **Since the introduction have you heard rumors?** | | | | | | | | | | |
| Yes | 65 | 84.4 | 52 | 66.7 | 51 | 65.4 | 66 | 43.4 | 42 | 52.5 |
| No | 11 | 14.3 | 25 | 32.1 | 27 | 34.6 | 85 | 55.9 | 38 | 47.5 |
| Don't know | 1 | 1.3 | 1 | 1.3 | 0 | 0.0 | 1 | 0.7 | 0 | 0.0 |
| **Common rumors*** | | | | | | | | | | |
| The vaccine affects fertility | 57 | 87.7 | 49 | 94.2 | 49 | 96.1 | 56 | 84.8 | 40 | 95.2 |
| The vaccine causes cancer | 4 | 6.2 | 2 | 3.8 | 0 | 0.0 | 0 | 0.0 | 0 | 0.0 |
| The vaccine has severe secondary effects | 6 | 9.2 | 5 | 9.6 | 5 | 9.8 | 9 | 13.6 | 2 | 4.8 |
| The vaccine costs money | 0 | 0.0 | 0 | 0.0 | 2 | 3.9 | 2 | 3.0 | 0 | 0.0 |
| The vaccine is not safe | 22 | 33.8 | 3 | 5.8 | 10 | 19.6 | 9 | 13.6 | 7 | 16.7 |
| The vaccine will promote early sexual activity | 0 | 0.0 | 1 | 1.9 | 0 | 0.0 | 1 | 1.5 | 1 | 2.4 |
| The vaccine is experimental | 2 | 3.1 | 1 | 1.9 | 2 | 3.9 | 0 | 0.0 | 2 | 4.8 |
| Other | 3 | 4.6 | 3 | 5.8 | 3 | 5.9 | 3 | 4.5 | 1 | 2.4 |
| **Origins of rumors+** | | | | | | | | | | |
| HCW | 6 | 11.8 | 4 | 8.9 | 6 | 15.8 | 1 | 2.4 | 3 | 8.6 |
| Social media/internet | 35 | 68.6 | 22 | 48.9 | 9 | 23.7 | 6 | 14.3 | 7 | 20.0 |
| Community | 8 | 15.7 | 18 | 40.0 | 14 | 36.8 | 27 | 64.3 | 18 | 51.4 |
| Teachers | 1 | 2.0 | 2 | 4.4 | 0 | 0.0 | 1 | 2.4 | 0 | 0.0 |
| Parents | 2 | 3.9 | 7 | 15.6 | 0 | 0.0 | 5 | 11.9 | 5 | 14.3 |
| Other | 0 | 0.0 | 2 | 4.4 | 0 | 0.0 | 2 | 4.8 | 4 | 11.4 |
| **Types of social mobilisation and communication materials received** | | | | | | | | | | |
| Training guide for HPV vaccination | 39 | 50.6 | 17 | 21.8 | 5 | 6.4 | 3 | 2.0 | 3 | 3.8 |
| EPI communication plan | 0 | 0.0 | 0 | 0.0 | 0 | 0.0 | 0 | 0.0 | 9 | 11.3 |
| Training slides | 0 | 0.0 | 0 | 0.0 | 1 | 1.3 | 0 | 0.0 | 0 | 0.0 |
| Brochures | 30 | 39.0 | 15 | 19.2 | 21 | 26.9 | 13 | 8.6 | 10 | 12.5 |
| Posters | 59 | 76.6 | 17 | 21.8 | 20 | 25.6 | 15 | 9.9 | 13 | 16.3 |
| Messages WhatsApp | 0 | 0.0 | 0 | 0.0 | 0 | 0.0 | 0 | 0.0 | 0 | 0.0 |
| No official materials | 7 | 9.1 | 43 | 55.1 | 42 | 53.8 | 120 | 78.9 | 54 | 67.5 |
| Other | 7 | 9.1 | 2 | 2.6 | 6 | 7.7 | 3 | 2.0 | 1 | 1.3 |
| Do not know | 4 | 5.2 | 4 | 5.1 | 3 | 3.8 | 7 | 4.6 | 6 | 7.5 |
| **In your opinion, what would be the three most effective channels of communication to reach those responsible for young girls with regard to HPV vaccination?** | | | | | | | | | | |
| Social media (Facebook, Whatsapp, Instagram etc.) | 67 | 87.0 | 8 | 10.3 | 9 | 11.5 | 10 | 6.6 | 8 | 10.0 |
| Radio | 39 | 50.6 | 21 | 26.9 | 29 | 37.2 | 37 | 24.3 | 21 | 26.3 |
| TV | 52 | 67.5 | 11 | 14.1 | 11 | 14.1 | 14 | 9.2 | 9 | 11.3 |
| Brochures | 75 | 97.4 | 3 | 3.8 | 5 | 6.4 | 3 | 2.0 | 3 | 3.8 |
| HCW | 65 | 84.4 | 20 | 25.6 | 17 | 21.8 | 53 | 34.9 | 23 | 28.8 |
| Community leaders | 60 | 77.9 | 31 | 39.7 | 33 | 42.3 | 56 | 36.8 | 37 | 46.3 |
| Community outreach workers/community sensitization | 51 | 66.2 | 34 | 43.6 | 64 | 82.1 | 114 | 75.0 | 65 | 81.3 |

(*Continued*)

**Table 4.** (Continued)

| | Healthcare Workers (N = 77) | | School Personnel (N = 78) | | Community Healthcare Workers (N = 78) | | Parents (N = 152) | | Community Leaders (N = 80) | |
|---|---|---|---|---|---|---|---|---|---|---|
| | **n** | **%** | **n** | **%** | **n** | **%** | **n** | **%** | **n** | **%** |
| Teachers or school officials | 63 | 81.8 | 49 | 62.8 | 19 | 24.4 | 25 | 16.4 | 9 | 11.3 |
| Other | 13 | 16.9 | 5 | 6.4 | 10 | 12.8 | 6 | 3.9 | 1 | 1.3 |
| Religious leaders | 7 | 9.1 | 13 | 16.7 | 10 | 12.8 | 16 | 10.5 | 17 | 21.3 |
| Parents | 0 | 0.0 | 5 | 6.4 | 2 | 2.6 | 2 | 1.3 | 2 | 2.5 |
| Don't know | 0 | 0.0 | 0 | 0.0 | 0 | 0.0 | 5 | 3.3 | 0 | 0.0 |
| **In your opinion, what would be the three most effective channels of communication to reach young girls with regard to HPV vaccination?** | | | | | | | | | | |
| Social Media (Facebook, Whatsapp, Instagram etc.) | 7 | 9.1 | 14 | 17.9 | 11 | 14.1 | 21 | 13.8 | 20 | 25.0 |
| Radio | 14 | 18.2 | 8 | 10.3 | 14 | 17.9 | 17 | 11.2 | 15 | 18.8 |
| TV | 19 | 24.7 | 4 | 5.1 | 9 | 11.5 | 17 | 11.2 | 11 | 13.8 |
| Brochures | 4 | 5.2 | 6 | 7.7 | 5 | 6.4 | 3 | 2.0 | 1 | 1.3 |
| HCW | 16 | 20.8 | 17 | 21.8 | 19 | 24.4 | 35 | 23.0 | 16 | 20.0 |
| Community leaders | 14 | 18.2 | 15 | 19.2 | 14 | 17.9 | 26 | 17.1 | 14 | 17.5 |
| Community outreach workers | 35 | 45.5 | 21 | 26.9 | 51 | 65.4 | 50 | 32.9 | 30 | 37.5 |
| Teachers or school officials | 49 | 63.6 | 62 | 79.5 | 51 | 65.4 | 87 | 57.2 | 43 | 53.8 |
| Other | 6 | 7.8 | 5 | 6.4 | 6 | 7.7 | 6 | 3.9 | 5 | 6.3 |
| Religious leaders | 2 | 2.6 | 11 | 14.1 | 15 | 19.2 | 6 | 3.9 | 8 | 10.0 |
| Parents | 29 | 37.7 | 21 | 26.9 | 12 | 15.4 | 15 | 9.9 | 5 | 6.3 |
| Youth groups associations | 3 | 3.9 | 2 | 2.6 | 1 | 1.3 | 10 | 6.6 | 4 | 5.0 |
| Do not know | 0 | 0.0 | 0 | 0.0 | 0 | 0.0 | 8 | 5.3 | 1 | 1.3 |
| **How comfortable are you recommending the vaccine?** | | | | | | | | | | |
| 5—Very comfortable | 34 | 44.2 | 29 | 37.2 | 36 | 46.2 | 40 | 26.3 | 33 | 41.3 |
| 4—A little comfortable | 21 | 27.3 | 34 | 43.6 | 22 | 28.2 | 47 | 30.9 | 20 | 25.0 |
| 3 –Neither comfortable nor uncomfortable | 20 | 26.0 | 6 | 7.7 | 18 | 23.1 | 36 | 23.7 | 17 | 21.3 |
| 2—A little uncomfortable | 1 | 1.3 | 3 | 3.8 | 2 | 2.6 | 13 | 8.6 | 7 | 8.8 |
| 1—Not at all comfortable | 1 | 1.3 | 6 | 7.7 | 0 | 0.0 | 16 | 10.5 | 3 | 3.8 |
| **Is the vaccine currently accepted in the community?** | | | | | | | | | | |
| 5 –Very accepted | 14 | 18.2 | 15 | 19.2 | 20 | 25.6 | 41 | 27.0 | 20 | 25.0 |
| 4—A little accepted | 37 | 48.1 | 32 | 41.0 | 40 | 51.3 | 72 | 47.4 | 42 | 52.5 |
| 3 –Neither accepted nor unaccepted | 17 | 22.1 | 16 | 20.5 | 13 | 16.7 | 33 | 21.7 | 12 | 15.0 |
| 2—A little unaccepted | 9 | 11.7 | 13 | 16.7 | 4 | 5.1 | 6 | 3.9 | 4 | 5.0 |
| 1—Not at all accepted | 0 | 0.0 | 2 | 2.6 | 1 | 1.3 | 0 | 0.0 | 2 | 2.5 |
| **Is there a demand for the vaccine?** | | | | | | | | | | |
| Yes | 59 | 76.6 | 37 | 47.4 | 64 | 82.1 | 101 | 66.4 | 54 | 67.5 |
| No | 18 | 23.4 | 25 | 32.1 | 13 | 16.7 | 27 | 17.8 | 15 | 18.8 |
| Don't know | 0 | 0.0 | 6 | 7.7 | 1 | 1.3 | 24 | 15.8 | 11 | 13.8 |
| **Is it important for young girls to receive the vaccine?** | | | | | | | | | | |
| Very important | - | - | 75 | 96.2 | - | - | 151 | 99.3 | 79 | 98.8 |
| Important enough | - | - | 1 | 1.3 | - | - | 1 | 0.7 | 1 | 1.3 |
| Not important | - | - | 0 | 0.0 | - | - | 0 | 0.0 | 0 | 0.0 |
| Do not know | - | - | 2 | 2.6 | - | - | 0 | 0.0 | 0 | 0.0 |

*among those indicating they had heard rumors

HPV = human papillomavirus.

Unreported values (-).

The most common rumor in the community reported by all respondent groups was that the HPV vaccine affects fertility (84.8–96.1%). Among the respondents, 33.8% of HCWs, 19.6% of cHCWs, 5.8% of the school personnel, 13.6% of the parents and 16.7% of the community leaders also reported hearing a rumor about the vaccine being unsafe. Other less frequently reported rumors included the vaccine having severe secondary effects, being experimental, and causing cancer. Over two-thirds of HCWs (68.6%) and almost half of the school personnel (48.9%) indicated that rumors originated from social media and the internet. About half of the parents (64.3%) and community leaders (51.4%) reported that the rumors originated from the community.

HCWs indicated that brochures (97.4%), social media (87.0%), and HCWs (84.4%) would be the most effective channels of communication to reach those responsible for young girls. School personnel felt that school officials (62.8%), community outreach workers (43.6%), and community leaders (39.7%) would be the most effective. CHCWs indicated that community outreach workers (82.1%), community leaders (42.3%), and the radio (37.2%) would be most effective. Parents and community leaders both indicated that community outreach workers (75.0% and 81.3%, respectively) community leaders (36.8% and 46.3%, respectively), and HCWs (34.9% and 28.8%, respectively) would be the most effective channels to reach those responsible for young girls. All respondents thought that the most effective communication channels for reaching young girls were teachers (53.8–79.5%) and community outreach workers (26.9–45.5%).

Over half of the respondents in each group felt very comfortable or comfortable recommending the HPV vaccine (HCWs: 72.0%, school personnel: 80.8%, cHCWs: 74.4%, parents: 57.2%, community leaders: 66.3%). Most of the respondents, 66.1% of the HCWs, 60.2% of the school personnel, 76.9% of the cHCWs, 74.4% of the parents, and 77.5% of the community leaders felt that the vaccine was very accepted or accepted in the community. The majority of HCWs, cHCWs, parents, and community leaders felt that there was a demand in the community (66.4–67.6%). Fewer than half of the school personnel felt that there was a demand in the community for the vaccine (47.4%). Almost all of the school personnel, parents, and community leaders (>95% in each group) felt it was important for young girls to receive this vaccine.

## Discussion

Senegal's experience illustrates how HPV vaccine delivery using routine delivery strategies, throughout the year, can be operationalized. Many countries introducing HPV vaccine have successfully utilized a nationwide, primarily school-based service delivery strategy, targeting young girls aged 9 or 10 years, once or twice a year [13, 16–19]. However, Senegal's utilization of the RI system and different sites for immunization service delivery (schools, community outreach and health facilities) with continuous immunization throughout the year is unique among early national programs in the African region.

As the HPV vaccine targets an age group not traditionally utilizing the childhood RI system, the integration of the HPV vaccine into the RI system requires some modifications [20]. Community healthcare workers, parents, and community leaders generally knew that vaccination was offered at both schools and health facilities. While respondents reported that there are still numerous 9-year-old girls that are out-of-school, only a few parents and community leaders knew about the availability of vaccination in the community. Similar to other countries, Senegal selected 9-year-old girls due to higher primary school enrollment, compared to secondary school enrollment [17–19, 21]. Moreover, Senegal's demonstration project resulted in high vaccination coverage (>90%) in a rural district when 37% of the girls were out of school [11]. However, Senegal and other countries with lower rates of school enrollment should consider

innovative ways of community outreach to access out-of-school girls that might be missed during school vaccinations, including mobilizing these girls to schools on vaccination days, encouraging health facility-based vaccination, or integrating outreach efforts [22–24]. A comprehensive approach of utilizing all routine delivery strategies is needed to achieve equitable high HPV vaccination coverage and has been shown to be successful in other countries [25, 26]. Overall, Senegal's HPV introduction was successful, with high-level political engagement, knowledgeable health staff, and high vaccine demand in communities. Nevertheless, the results of this evaluation demonstrate that continued government efforts are needed to ensure complete integration into the RI system. Success is dependent on a multisectoral approach with strong partnerships between the education and health sectors [22].

More than half of HCWs reported that the HPV vaccine introduction had an impact on their workload, which is an important consideration because this could weaken HCW commitment if staff feel overburdened. In turn, RI activities could also suffer over time [27]. To ensure the sustainability of the HPV vaccination as well as other RI activities, programs should confirm the allocation of sufficient resources, including staffing and funds to cover additional activities required for HPV vaccination. Senegal could also consider a reallocation of funds within the immunization budget to account for the increased effort needed to deliver the HPV vaccine in schools and to assure the sustainability of the routine delivery approach, including school-based vaccination.

Overall, this evaluation demonstrated that the training and orientation efforts were highly successful in developing a good understanding of standard HPV vaccine and vaccination concepts among all groups. However, some areas may need reinforcement among HCWs, particularly around contraindications for the HPV vaccination. Considering the high staff turnover, continuous provision of supportive supervision, frequent trainings, and refreshers will be necessary for both existing and new HCWs and school personnel [28, 29]. Additionally, since conducting this evaluation, the COVID-19 pandemic has led to disruptions in RI services; therefore, staff will need training on catch-up vaccination. Ensuring that healthcare workers are knowledgeable can lead to improvements in health communication messaging [30].

Senegal's successful communication and social mobilization efforts created high acceptance and demand for the HPV vaccine. However, there were challenges relating to rumors circulating within the community prior to and during HPV introduction. The most common rumors were about vaccine safety and the risk of infertility, which have been documented elsewhere [31]. Senegal responded well to these rumors in the first year. A committee was formed with a Ministry of Health and Social Action spokesperson and a specialized communication plan was developed with focused efforts on crisis communication; updates were made to a cervical cancer video that included the HPV vaccine, and experts provided key messages in Wolof, the local language. Circulating rumors should not be taken lightly, as gender-specific immunization has been historically challenging in many cultures [32]. Rumors and vaccine hesitancy can spread quickly, particularly with current social media channels, that can rapidly destroy RI programs. Rebuilding community trust in vaccination and other public health interventions can be quite resource-intensive and may take many years [33]. While preparing for the HPV vaccine introduction, countries should develop crisis management strategies and train the community stakeholders with reliable information, including facts, key messages, and how previous rumors were addressed. Additionally, HCWs indicated inadequate availability of social mobilization materials. Community leaders and parents often reported that they had not received any materials prior to the introduction of HPV vaccine. Because the HPV vaccination program is new and the rumors will likely continue to circulate, it is necessary to have ongoing social mobilization and sensitization efforts to build lasting community trust. Reprinting and redistribution of print materials should be ongoing. Additionally, empowering

and equipping community leaders could improve vaccination coverage and overall community acceptance and reduce the circulation of rumors [34, 35]. In Sierra Leone, religious leader engagement and partnerships with the media were central to the social mobilization strategy [36]. In India, in areas at risk for polio, mass media, print materials, house-to-house dialogs, peer-support groups and the training of traditional and religious leaders were associated with improved coverage [37]. Senegal should also consider the development of specific tools to combat the most common rumors around fertility and vaccine safety.

This evaluation is subject to several limitations. While the sampling of health facilities was intended to be nationally representative, the individuals selected to interview at those health facilities represent a convenience sample; hence the results cannot be extrapolated to all community stakeholders. Data collection occurred more than a year after the national HPV vaccine introduction rollout and the respondents may not have had an accurate recall of all information. We attempted to interview staff involved in the HPV vaccination program; however, given the high staff turnover, not all respondents would have been present for the rollout in October 2018 and may had have varying levels of experience with the program. Each questionnaire was translated, piloted, and adapted to the country-context, and interviewers received special training, however, some questions and response choices may have lacked clarity and led to errors in translation.

This program evaluation aimed to better understand Senegal's HPV vaccine introduction experience by providing a snapshot of the feasibility and acceptability of the program and identify areas of improvement. As 16 Gavi-eligible countries plan HPV vaccine introductions by 2022 [38, 39], Senegal's experience should provide valuable lessons for adaptation of an RI delivery strategy to an older age group. As the HPV vaccination program matures in Senegal, continued support will be necessary to ensure the success and sustainability of the program. Continuing to strengthen the HPV vaccination program by equipping all individuals with HPV vaccine knowledge, supporting health facilities in outreach planning, and continuing to raise awareness around the HPV program is essential for sustainability.

## Acknowledgments

**Disclaimer:** The findings and conclusions expressed in this article are those of the author(s) and do not necessarily represent the views, decisions, or policies of the institutions with which the authors are affiliated.

## Author Contributions

**Conceptualization:** Rebecca M. Casey, Alassane Ndiaye, Jerlie Loko Roka, Julie Garon, Anagha Loharikar.

**Data curation:** Rebecca M. Casey, Nedghie Adrien, Alassane Ndiaye.

**Formal analysis:** Reena H. Doshi, Rebecca M. Casey, Nedghie Adrien.

**Methodology:** Alassane Ndiaye, Jerlie Loko Roka, Awa Bathily, Anyie Li, Julie Garon, Ousseynou Badiane, Aliou Diallo, Anagha Loharikar.

**Project administration:** Reena H. Doshi, Alassane Ndiaye, Timothy Brennan, Jerlie Loko Roka, Awa Bathily, Cathy Ndiaye, Anyie Li, Ousseynou Badiane, Anagha Loharikar.

**Resources:** Anagha Loharikar.

**Supervision:** Reena H. Doshi, Timothy Brennan, Jerlie Loko Roka, Awa Bathily, Cathy Ndiaye, Julie Garon, Ousseynou Badiane, Aliou Diallo, Anagha Loharikar.

**Validation:** Anagha Loharikar.

**Writing – original draft:** Reena H. Doshi, Rebecca M. Casey, Nedghie Adrien, Anyie Li, Anagha Loharikar.

**Writing – review & editing:** Reena H. Doshi, Rebecca M. Casey, Nedghie Adrien, Alassane Ndiaye, Timothy Brennan, Jerlie Loko Roka, Awa Bathily, Cathy Ndiaye, Anyie Li, Julie Garon, Ousseynou Badiane, Aliou Diallo, Anagha Loharikar.

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
