## [Decision Letter · Decision Letter 0]

13 Oct 2021

PGPH-D-21-00369

Feasibility and acceptability of nationwide HPV vaccine introduction in Senegal: findings from community-level cross-sectional surveys, 2020

Dear Dr. Doshi,

Thank you for submitting your manuscript to PLOS Global Public Health. After careful consideration, we feel that it has merit but does not fully meet PLOS Global Public Health’s publication criteria as it currently stands. Therefore, we invite you to submit a revised version of the manuscript that addresses the points raised during the review process.

Thanks for submitting a good manuscript on a relevant topic. There are some minor comments to be addressed before a final decision can be reached.

We look forward to receiving your revised manuscript.

Kind regards,

Charles Anawo Ameh

Academic Editor

Journal Requirements:

1. We note that participants provided oral consent. Please state in the Methods:

- Why written consent could not be obtained

- How oral consent was documented

For more information, please see our guidelines for human subjects research: https://journals.plos.org/plosone/s/submission-guidelines#loc-human-subjects-research

2. You indicated that ethical approval was not necessary for your study. We understand that the framework for ethical oversight requirements for studies of this type may differ depending on the setting and we would appreciate some further clarification regarding your research. Could you please provide further details on why your study is exempt from the need for approval and confirmation from your institutional review board or research ethics committee (e.g., in the form of a letter or email correspondence) that ethics review was not necessary for this study? Please include a copy of the correspondence as an ""Other"" file.

3. Please provide separate figure files in .tif or .eps format only, and remove any figures embedded in your manuscript file.  If you are using LaTeX, you do not need to remove embedded figures.

4. We do not publish any copyright or trademark symbols that usually accompany proprietary names, eg (R), (C), or TM  (e.g. next to drug or reagent names). Therefore please remove all instances of trademark/copyright symbols throughout the text, including Microsoft Excel® on page 6.

5. Please update the completed 'Competing Interests' statement, including any COIs declared by your co-authors. If you have no competing interests to declare, please state "The authors have declared that no competing interests exist". Otherwise please declare all competing interests beginning with the statement "I have read the journal's policy and the authors of this manuscript have the following competing interests:"

6. Since your data is not available for proprietary reasons, please explain via email why the data is not available. Please also include the contact information for the third party organization that should be contacted should other researchers want to request access to this data and please include the full citation of where the data can be found. We also request that you verify with us via email that any researcher will be able to obtain the data set in the same manner that the you have obtained it. If you feel you are unwilling or unable to adhere to this policy, please explain your reasons by return email and your exemption request will be escalated to the editor for approval. Your exemption request will be handled independently and will not hold up the peer review process, but will need to be resolved should your manuscript be accepted for publication. One of the Editorial team will be in touch if they require more information.

7. Please amend your detailed Financial Disclosure statement. This is published with the article, therefore should be completed in full sentences and contain the exact wording you wish to be published.

i). State the initials, alongside each funding source, of each author to receive each grant.

ii). State what role the funders took in the study. If the funders had no role in your study, please state: “The funders had no role in study design, data collection and analysis, decision to publish, or preparation of the manuscript.”

Reviewers' comments:

Reviewer's Responses to Questions

**Comments to the Author**

1. Does this manuscript meet PLOS Global Public Health’s publication criteria? Is the manuscript technically sound, and do the data support the conclusions? The manuscript must describe methodologically and ethically rigorous research with conclusions that are appropriately drawn based on the data presented.

Reviewer #1: Yes

Reviewer #2: Yes

2. Has the statistical analysis been performed appropriately and rigorously?

Reviewer #1: Yes

Reviewer #2: Yes

3. Have the authors made all data underlying the findings in their manuscript fully available (please refer to the Data Availability Statement at the start of the manuscript PDF file)?

Reviewer #1: Yes

Reviewer #2: Yes

4. Is the manuscript presented in an intelligible fashion and written in standard English?

Reviewer #1: Yes

Reviewer #2: Yes

5. Review Comments to the Author

Reviewer #1: Thank you for your interesting article on a very important topic.

The data presented is extensive and provides useful information that can inform ongoing implementation efforts for the national HPV vaccination program in Senegal.

There are a few issues that I suggest need to be addressed in order to strengthen the article (please see the attached document).

Reviewer #2: This is a clearly written paper reflecting a well done study on an important topic for the global immunization and cancer control communities. Cervical cancer, the number one or two cancer killer of women in much of the developing world is highly preventable with safe and very effective vaccines when given to girls before the onset of sexual activity. GAVI’s ability to procure affordable HPV vaccines and supply them to the poorest countries represents great progress in global cancer control, and the success of the pilot and early national program in Senegal is very encouraging and important.

This study of knowledge and practices of HPV immunization in randomly selected communities in Senegal is well conceived, rigorously conducted, properly analyzed, and intelligently discussed. The discussion also provides useful advice for immunization programs in future GAVI countries and beyond. I have only a few comments and suggestions.

I think the introduction could be strengthened with some additional information such as Senegal’s immunization coverage with measles and DTP vaccines; data, if available yet, on National HPV coverage from WHO/UNICEF data or national surveys; and UNICEF estimates on the proportion of nine year old girls in school. The survey asks HCW and school personnel to estimates how many 9 year olds attend school, but they don’t know. Perhaps outside data may help to clarify this.

I was initially unclear what was meant by “community” delivery of the vaccine. Is outreach to girls who don’t attend school or who missed a dose at school all of it, or are there other mechanisms of community delivery?

It is mentioned in the methods section and discussion that although the communities studied are randomly chosen, from that point on this is a convenience sample. The interviewee school staff and parents were chosen by a local HCW who would likely choose a school and individuals he or she knew would understand the program well. Why was this method of selection chosen? The schools and parents could have been randomly chosen. A brief discussion of this would be useful, and mentioning that this is a convenience sample in the abstract might also be appropriate.

Overall, an important and well done study.

6. PLOS authors have the option to publish the peer review history of their article (what does this mean?). If published, this will include your full peer review and any attached files.

**Do you want your identity to be public for this peer review?** For information about this choice, including consent withdrawal, please see our Privacy Policy.

Reviewer #1: No

Reviewer #2: No

---

## [Decision Letter · Decision Letter 1]

2 Feb 2022

Feasibility and acceptability of nationwide HPV vaccine introduction in Senegal: findings from community-level cross-sectional surveys, 2020

PGPH-D-21-00369R1

Dear Dr. Doshi,

We are pleased to inform you that your manuscript 'Feasibility and acceptability of nationwide HPV vaccine introduction in Senegal: findings from community-level cross-sectional surveys, 2020' has been provisionally accepted for publication in PLOS Global Public Health.

Best regards,

Charles Anawo Ameh, PhD

Academic Editor

Thanks for addressing all the comments from the reviewers, I am please to make a recommendation of 'accept'. Congratulations.

Reviewer Comments (if any, and for reference):

Reviewer's Responses to Questions

**Comments to the Author**

1. If the authors have adequately addressed your comments raised in a previous round of review and you feel that this manuscript is now acceptable for publication, you may indicate that here to bypass the “Comments to the Author” section, enter your conflict of interest statement in the “Confidential to Editor” section, and submit your "Accept" recommendation.

Reviewer #1: All comments have been addressed

Reviewer #2: All comments have been addressed

2. Does this manuscript meet PLOS Global Public Health’s publication criteria? Is the manuscript technically sound, and do the data support the conclusions? The manuscript must describe methodologically and ethically rigorous research with conclusions that are appropriately drawn based on the data presented.

Reviewer #1: Yes

Reviewer #2: (No Response)

3. Has the statistical analysis been performed appropriately and rigorously?

Reviewer #1: Yes

Reviewer #2: (No Response)

4. Have the authors made all data underlying the findings in their manuscript fully available (please refer to the Data Availability Statement at the start of the manuscript PDF file)?

Reviewer #1: Yes

Reviewer #2: (No Response)

5. Is the manuscript presented in an intelligible fashion and written in standard English?

Reviewer #1: Yes

Reviewer #2: (No Response)

6. Review Comments to the Author

Reviewer #1: The authors have addressed all comments satisfactorily.

This is a well written paper on an important subject and I recommend that it be pushed in PLOS Global Public Health.

Reviewer #2: (No Response)

7. PLOS authors have the option to publish the peer review history of their article (what does this mean?). If published, this will include your full peer review and any attached files.

**Do you want your identity to be public for this peer review?** For information about this choice, including consent withdrawal, please see our Privacy Policy.

Reviewer #1: No

Reviewer #2: No
